# Structure of the SARS-CoV nsp12 polymerase bound to nsp7 and nsp8 co-factors

Robert N. Kirchdoerfer [1] & Andrew B. Ward [1]

Recent history is punctuated by the emergence of highly pathogenic coronaviruses such as SARS- and MERS-CoV into human circulation. Upon infecting host cells, coronaviruses assemble a multi-subunit RNA-synthesis complex of viral non-structural proteins (nsp) responsible for the replication and transcription of the viral genome. Here, we present the 3.1 Å resolution structure of the SARS-CoV nsp12 polymerase bound to its essential co-factors, nsp7 and nsp8, using single particle cryo-electron microscopy. nsp12 possesses an architecture common to all viral polymerases as well as a large N-terminal extension containing a kinase-like fold and is bound by two nsp8 co-factors. This structure illuminates the assembly of the coronavirus core RNA-synthesis machinery, provides key insights into nsp12 polymerase catalysis and fidelity and acts as a template for the design of novel antiviral therapeutics.

[1] Department of Integrative Structural and Computational Biology, The Scripps Research Institute, 10550 North Torrey Pines Road, HZ-102, La Jolla, CA 92037, USA. Correspondence and requests for materials should be addressed to R.N.K. (email: rkirchdo@scripps.edu)

Coronaviruses (CoV) represent a diverse family of positive-sense RNA viruses capable of causing respiratory and enteric disease in human and animal hosts. Though there are several human CoV responsible for a mild respiratory disease[1], most notable are the highly pathogenic human CoVs: SARS-CoV and MERS-CoV capable of causing a severe respiratory disease. The zoonotic SARS-CoV emerged into human populations in 2002, spreading to 26 countries during its brief 9-month circulation in humans[2]. This epidemic resulted in >8000 infections with a ~10% case fatality rate and was eventually contained through public health measures, as there are no specific treatments approved for human CoV infections. With the exception of a smaller second SARS-CoV outbreak in 2004[3], SARS-CoV has been absent from human circulation since the initial outbreak ended. Despite the lack of recent human infections, SARS-CoV-like viruses continue to circulate in bat reservoirs[4]. Outbreaks of highly pathogenic human CoVs remain an emerging threat to global health security and are likely to continue to occur.

The threat of a novel emerging virus from the diverse CoV family necessitates antiviral strategies targeting conserved elements of the viral life cycle such as the viral machinery responsible for the replication and transcription of the positive-strand viral RNA genome. The multi-subunit CoV RNA synthesis machinery is a complex of non-structural proteins (nsp) produced as cleavage products of the ORF1a and ORF1ab viral polyproteins[5]. The nsp12 RNA-dependent RNA polymerase possesses some minimal activity on its own[6], but the addition of the nsp7 and nsp8 co-factors greatly stimulates polymerase activity[7]. Though additional viral nsp subunits are likely necessary to carry out the full repertoire of replication and transcription activities, the nsp12-nsp7-nsp8 complex so far represents the minimal complex required for nucleotide polymerization. In addition, many other CoV nsps possess enzyme activities related to RNA modification[8,9].

After the emergence of SARS-CoV, there was an intense effort to structurally characterize the replication complexes of CoVs. This resulted in high-resolution structure determination for many of the SARS-CoV nsps in isolation using X-ray crystallography and nuclear magnetic resonance[10,11]. In addition, complexes of nsp7-nsp8, nsp10-nsp14 and nsp10-nsp16 were also determined[12–14]. One notable gap in our structural knowledge of the CoV RNA synthesis complex was a structure of the nsp12 RNA polymerase. This structural gap included information regarding the unique N-terminal extension of nidovirus polymerases, which has been proposed to contain a nucleotidyltransferase activity (nidovirus RdRp-associated nucleotidyltransferase (NiRAN))[9]. It had also been unclear what the role of nsp8 and the nsp7-nsp8 complex within the viral RNA synthesis complex play, with proposed functions ranging from acting as processivity factors[14] to possessing RNA primase activity[15,16].

Here we used cryo-electron microscopy (cryo-EM) to determine the 3.1 Å resolution structure of SARS-CoV nsp12 bound to the nsp7 and nsp8 co-factors. This 160 kDa complex defines the largest portion of the viral RNA synthesis complex yet to be structurally characterized at high resolution. In addition to a structurally conserved polymerase domain, this complex provides the first views of a nidovirus-unique nsp12 N-terminal extension that possesses a kinase-like fold and demonstrates an unexpected binding stoichiometry of nsp7 and nsp8 co-factors.

## Results

### Structure description

Our cryo-EM structure shows the nsp12 polymerase bound to an nsp7-nsp8 heterodimer with a second subunit of nsp8 occupying a distinct binding site (Fig. 1a, b).

nsp12 contains a polymerase domain (a.a. 398–919) that assumes a structure resembling a cupped "right hand" similar to other polymerases[17] (Fig. 1c). The polymerase domain is comprised of a fingers domain (amino acids (a.a.) 398–581, 628–687), a palm domain (a.a. 582–627, 688–815), and a thumb domain (a.a. 816–919). CoV nsp12 also contains a nidovirus-unique N-terminal extension (a.a. 1–397) of which a.a. 117–397 were resolved in the cryo-EM map (Supplementary Figs. 1 and 2 and Supplementary Table 1).

Although not previously predicted, the SARS-CoV nsp12 contains two metal-binding sites to which we have assigned zinc atoms. The first is in the nidovirus-unique extension and is coordinated by His295, Cys301, Cys306, and Cys310. The second is in the fingers domain and is coordinated by Cys487, His642, Cys645, and Cys646. All eight of these metal-coordinating a.a. are highly conserved across the CoV family (Supplementary Fig. 3). Both of these metal-binding sites are distal to known active sites as well as protein–protein and protein–RNA interactions. Thus these ions are expected to be structural components of the folded protein rather than directly involved in enzymatic activity. The presence of structural zinc ions in nsp12 is reminiscent of bound zinc atoms in coronavirus nsp3, nsp10, nsp13, and nsp14[10,13,18,19] and points to an extensive utilization of zinc ions for folding proteins of the viral replication complex.

The outer surface of nsp12 carries a largely negative electrostatic potential (Fig. 2a). However, the polymerase RNA template and nucleotide triphosphate (NTP)-binding sites carry a strong positive electrostatic potential. The nsp7 and two nsp8-binding sites as well as the RNA exit tunnel are comparatively neutral. Similarly, the nsp7 and nsp8 surfaces contacting nsp12 are also relatively neutral. The second subunit of nsp8 contains some basic residues in the N-terminal region visible in the structure (a.a. 77–98) contributing to an extension of the positive electrostatics of the template-binding channel.

Analysis of sequence conservation across the CoV family reveals that the template entry, template-primer exit, and NTP tunnels, as well as the polymerase-active site, are the most highly conserved surfaces on nsp12 (Fig. 2b). There is also a conserved surface on the nsp12 nidovirus-unique extension, which may represent an interaction site for the N-terminal disordered domain of nsp12 (1–116) that is not resolved in the structure (Fig. 2c, d). The binding site for the nsp7-nsp8 heterodimer is well conserved and overlaps with the conserved regions of polymerase functional domains (fingers and thumb domains). Conversely, there is less conservation of the second nsp8-binding site on nsp12. More conserved nsp12 residues contact the nsp8 N-terminal region (77–126), while contacts with the nsp8 C-terminal head domain are mediated primarily by main chain atoms, which may have less stringent requirements in their a.a. composition to retain binding.

### nsp12 N-terminal extension

The SARS-CoV nsp12 is 932 a.a. in length, which is in contrast to the polymerases of the closely related picornaviruses that are typically closer to 500 a.a. While the C-terminal region of nsp12 contains the RNA polymerase domain, the large N-terminal nidovirus-unique extension has long remained enigmatic. Sequence analysis of polymerase subunits across the *Nidovirales* order, which includes CoVs, revealed three conserved sequence motifs within this region: $A_N$ (SARS-CoV a.a. 69–89), $B_N$ (a.a. 112–130), and $C_N$ (a.a. 202–222)[9] (Supplementary Fig. 3). Further experimentation revealed that the equine arterivirus polymerase subunit could be covalently modified with GTP or UTP and that mutations within the identified conserved motifs had negative impacts on this activity as well as viral growth and recovery, the proposed active site being a lysine

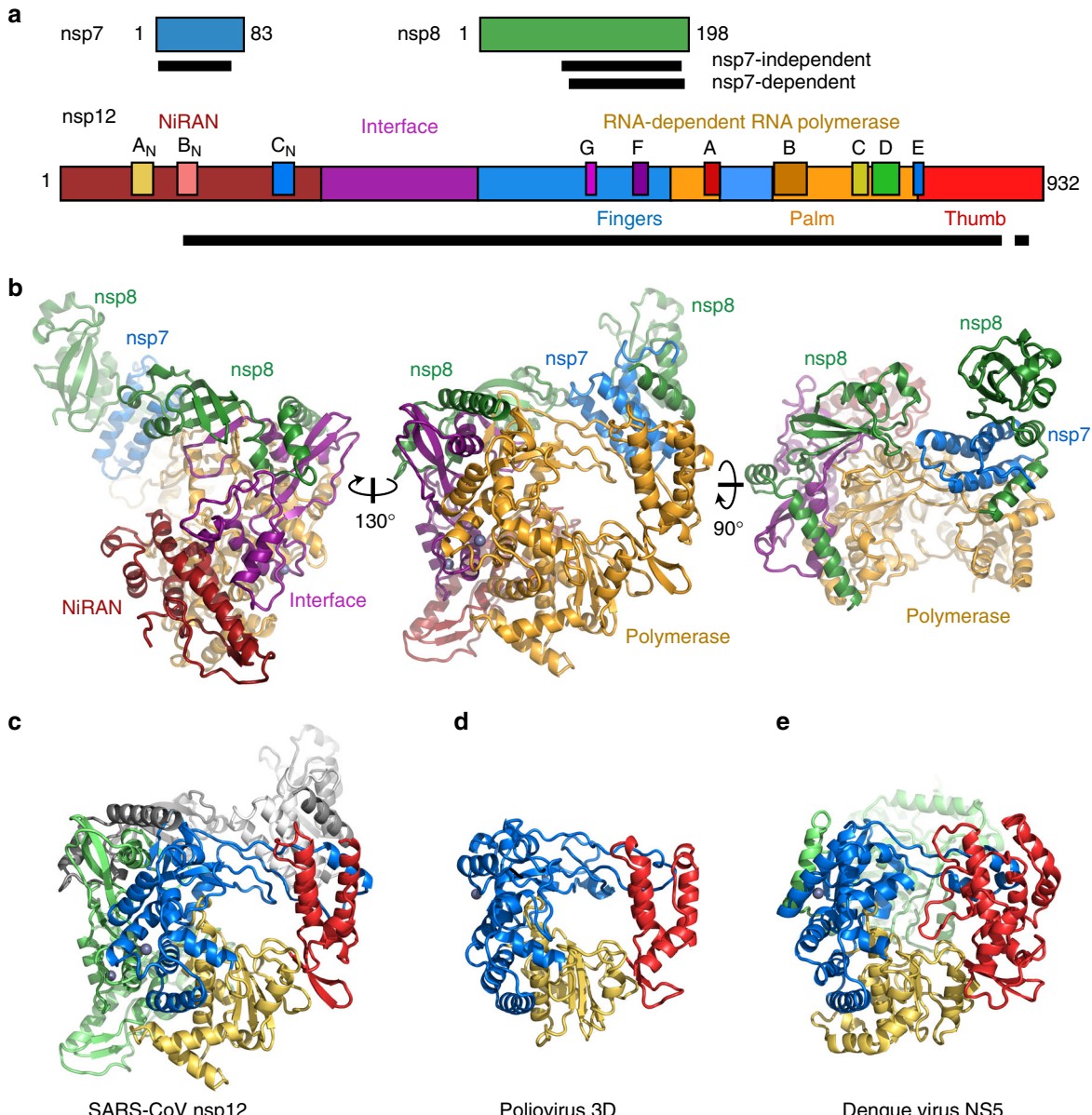

**Fig. 1** Structure of SARS-CoV nsp12 bound to nsp7 and nsp8 co-factors. **a** Diagram of the SARS-CoV nsp7, nsp8, and nsp12 proteins indicating domains, conserved motifs, and the protein regions observed in the structure (black bars) (Supplementary Fig. 3). **b** SARS-CoV nsp12 contains a large N-terminal extension composed of the NiRAN domain (dark red) and an interface domain (purple) adjacent to the polymerase domain (orange). nsp12 binds to a heterodimer of nsp7 (blue) and nsp8 (green) as well as to a second subunit of nsp8. **c–e** Comparison of SARS-CoV nsp12 to the polymerase proteins of poliovirus[24] (3OL6.pdb) [10.2210/pdb3OL6/pdb] and dengue virus[55] (4V0R.pdb) [10.2210/pdb4V0R/pdb]. Viral RNA polymerases share an overall structural architecture with fingers (blue), palm (yellow), and thumb domains (red). Both SARS-CoV and dengue virus polymerase proteins contain N-terminal extensions colored green. The SARS-CoV nsp7 and nsp8 cofactors are colored in white and gray, respectively

residue in $A_N$ (SARS-CoV Lys73)[9]. The conserved regions of the polymerase harboring this nucleotidyltransferase activity were termed NiRAN[9]. However, the role and mechanisms of this nucleotidyltransferase activity remain to be elucidated.

The nsp12 nidovirus-unique extension contacts the polymerase on the outside of the fingers domain and at the base of the palm domain (Fig. 1). Despite attempts at focused refinement of the cryo-EM map, the N-terminal 116 a.a. of nsp12 are not visible in the structure. Diffuse density localized to this region in the cryo-EM reconstructions suggests that the N-terminus is present but highly flexible. The portion of the nsp12 N-terminal extension that we observe in the map (a.a. 117–397) forms a contiguous domain with the fingers domain using a large number of well-conserved hydrophobic interactions.

The nidovirus-unique extension observed in the structure can be divided into at least two distinct regions: the NiRAN domain (a.a. 117–250) and an Interface domain (251–398) though both regions are likely to be reliant on one another and on the polymerase domains for proper folding, hence domain boundaries are ambiguous. The interface domain (a.a. 251–398) acts as a protein-interaction junction, contacting the NiRAN domain, the fingers domain, and the second subunit of nsp8.

The observed NiRAN domain contains the majority of motif $B_N$ as well as motif $C_N$. This region interacts with the base of the polymerase palm and fingers domains. It is interesting that the observed ordered region in the structure begins within the conserved sequence motif $B_N$ and implies a functional role for this linker region between the N-terminal disordered domain and

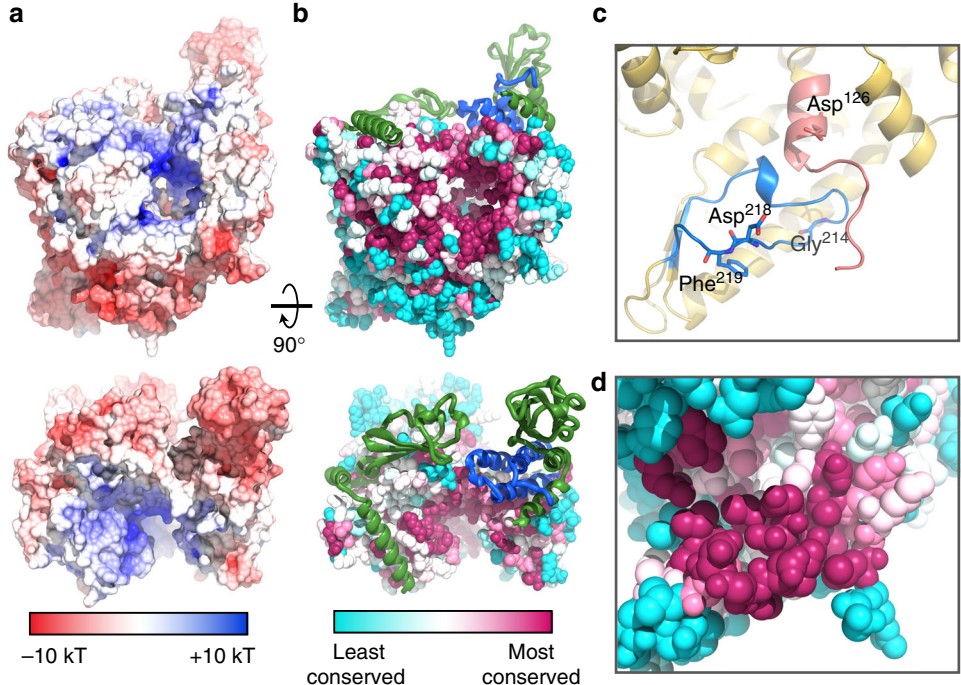

**Fig. 2** Surface electrostatics and sequence conservation of the nsp7-nsp8-nsp12 complex. **a** The RNA-binding cleft of SARS-CoV nsp7-nsp8-nsp12 is positively charged while the remaining protein surface carries a negative charge. **b** Sequence conservation[53,54] across the coronavirus family indicates a high level of conservation in the polymerase-active site and nsp7-nsp8 heterodimer-binding site with the binding site of the single nsp8 subunit less conserved. **c, d** Sequence motifs of the NiRAN domain[9] form a single surface on nsp12 (yellow). Motif $C_N$ (blue) and the C-terminal portion of motif $B_N$ (pink) are observed in the structure. Amino acids conserved across the *Nidovirales* order are shown as sticks. Coloring in **d** is the same as in **b**

the remaining nsp12 domains. The conserved a.a. from motifs $B_N$ and $C_N$ appear to form a single site on nsp12 (Fig. 2c, d). Of the seven NiRAN a.a. that are strictly conserved across the *Nidovirales* order, we observe four. Asp126 forms the center of the conserved face while Asp218 and F219 are on an external loop near the bottom of the conserved face. While these three a.a. are surface exposed, Gly214 is in a loop turn and is directed away from the conserved face possibly indicating this residue's importance in proper positioning of the downstream Asp218 and F219 rather than direct involvement in NiRAN activity.

**nsp12 RNA-dependent RNA polymerase**. As noted above, the SARS-CoV RNA-dependent RNA polymerase resembles a cupped right hand with fingers, palm, and thumb subdomains. Within the fingers domain are the index, middle, ring, and pinky finger loops. In positive-strand RNA virus polymerases, the index finger loop reaches over the active site to contact the thumb domain, positioning the ring-finger loop beneath it[20]. In SARS-CoV nsp12, the contacts between the index finger and the thumb domain are particularly extensive with the positioning of an alpha-helix in the index finger loop to pack with the thumb helical bundle. The index finger–thumb interaction site also forms the nsp7-nsp8 heterodimer-binding site, with most of the contacts made between nsp12 and nsp7.

All viral polymerases possess seven conserved motif regions (A–G) involved in template and nucleotide binding and catalysis[21–23] (Figs. 1a and 3a and Supplementary Fig. 3). Superposition of the elongation complexes of poliovirus[24] and norovirus[25] polymerases onto the SARS-CoV nsp12 reveals the path of the RNA past the active site and indicates a.a. involved in binding or catalysis (Fig. 3b). Single-stranded RNA template threads its way past motif G before entering the active site composed of motifs A and C and supported by motifs B and D.

Incoming NTPs would enter the active site through a tunnel at the back and interact with motif F. Motif E at the base of the thumb interacts with the 3′ nucleotide of the primer strand[26]. The primer-template product of RNA synthesis exits the active site though the RNA exit tunnel. This double-stranded RNA product would interact with the N-terminal region of motif G in the fingers domain via the major groove while a helix (SARS-CoV a.a. 851–864) from the thumb domain interacts with the minor groove.

CoVs possess the largest known RNA genomes and require an RNA synthesis complex with the fidelity to faithfully replicate their RNA. In the polymerase-active site, incoming NTPs form a base pair with the template RNA while the 2′ and 3′ hydroxyls form hydrogen bonds with the polymerase. In SARS-CoV nsp12, the 2' hydroxyl of the incoming NTP is likely to form hydrogen bonds with Thr680 and Asn691 in motif B. In addition, Asp623 in motif A is positioned to interrogate the 3′ hydroxyl though hydrogen bonding. All three of these residues are conserved across the CoV family.

Base pairing of the incoming NTP is also facilitated by stacking of the +1 template RNA base on a hydrophobic side chain in motif F. In SARS-CoV nsp12, this a.a. is Val557. The nucleoside analog GS-5734 is capable of impairing CoV RNA synthesis by targeting the viral RNA synthesis machinery[27]. When murine hepatitis virus, a *Betacoronavirus* related to SARS-CoV, was passaged in the presence of GS-5734, two mutations in nsp12 with partial resistance arose: Phe480Leu and Val557Leu[27] (SARS-CoV nsp12 numbering). The presence of the bulkier leucine side chain at a.a. 557 is likely to create a greater stringency for base pairing to the templating nucleotide possibly enabling nsp12 to better exclude this analog from its active site. The 557 a.a. position has also previously been indicated in modulating polymerase fidelity[28]. A.a. 480 is located in the fingers domain contacting motif B and may impact active-site dynamics related to catalysis.

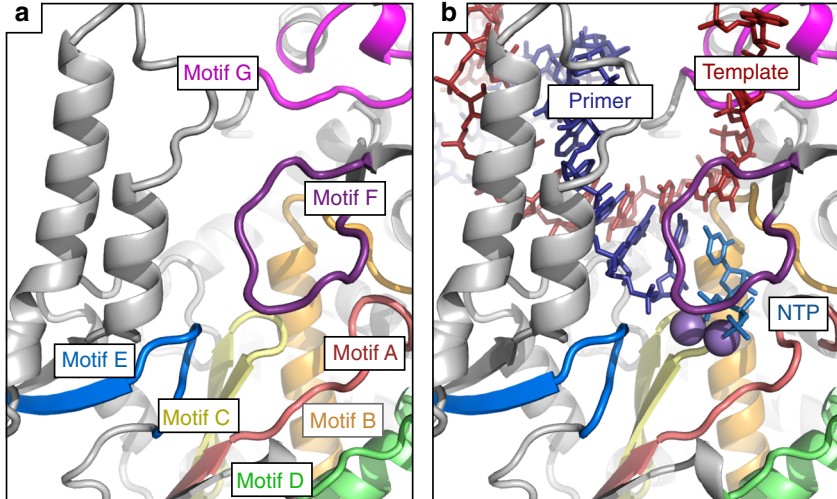

**Fig. 3** Architecture of the nsp12 polymerase-active site. **a** RNA polymerases possess seven conserved motifs (motifs A–G) involved in template binding (motif G), NTP binding (motif F), and polymerization (motifs A–E). **b** Superposition of elongation complexes from poliovirus[24] (3OL6.pdb) [10.2210/pdb3OL6/pdb] and norovirus[25] (3H5Y.pdb) [10.2210/pdb3H5Y/pdb] polymerases give the approximate positions of RNA template (red), primer (dark blue), incoming NTP (blue), and bound catalytic metal ions (purple)

**Interactions of nsp12 with nsp7 and nsp8.** The CoV RNA synthesis complex is governed by a large number of protein–protein interactions. Both nsp7 and nsp8 have essential roles in the formation and activity of the RNA synthesis machinery[7] (Supplementary Fig. 4). The co-crystal structure of SARS-CoV nsp7 and nsp8 demonstrated that the nsp8 C-terminal head domain folds around the nsp7 helical bundle[14]. The N-terminal region of nsp8 (1–81) adopts a more extended or disordered conformation. In the crystal packing of the nsp7-nsp8 structure, the authors hypothesized the formation of a hetero-hexadecamer protein complex. However, for feline CoV, the co-crystal structure of nsp7 and nsp8 displayed a 2:1 stoichiometry and lacked the higher-order oligomer formation hypothesized in the SARS-CoV nsp7-nsp8 structure[29]. nsp8 has also been suggested to possess an activity polymerizing short RNA oligonucleotides proposed to be used as primers during RNA synthesis[15]. Mutagenesis has suggested that this primase activity lies in the N-terminal region of nsp8 and requires the formation of large oligomeric complexes to bring the active site residues into proximity[15,16]. However, more recent work has shown that the SARS-CoV nsp7-nsp8-nsp12 complex was capable of de novo initiation and that this activity was dependent on the nsp12 polymerase-active site[7].

A number of studies have confirmed the direct interaction between nsp8 and nsp12[30–32]. In addition, nsp12 has also been suggested to interact with nsp5, nsp9, and nsp13[30] while nsp8 has been suggested to interact with nsp7, nsp8, nsp9, nsp10, nsp13, and nsp14[32]. The large number of interactions between nsp8 and nsp12 with the other viral nsps suggests that these two proteins form a hub for protein–protein interactions within the viral replication complex. However, the structures of nsp7, nsp8, and nsp12 in complex, their stoichiometry, and how they present their surfaces for additional interactions had remained unclear.

Examination of the cryo-EM map revealed density for a single complex of nsp7-nsp8 heterodimer bound to nsp12 resembling previously determined crystal structures (Fig. 4a). This density includes nsp7 a.a. 2–71 and nsp8 a.a. 84–192 encompassing the nsp8 head domain while the N-terminal region of nsp8 appears to be disordered. The nsp7-nsp8 heterodimer binds to nsp12 on the polymerase thumb domain facing the NTP entry channel. Binding in this position sandwiches the nsp12 polymerase index finger loop between nsp7-nsp8 and the polymerase thumb domain. The nsp12 index finger loop has been previously

identified as necessary for recruitment of nsp12 to replication complexes[30]. The binding of the nsp7-nsp8 heterodimer to this loop suggests that nsp7-nsp8 facilitates the interaction of nsp12 with additional components of the RNA synthesis machinery for incorporation into viral replication complexes.

Although nsp7 and nsp8 are known to form a 1:1 complex[14] and were provided in a 1:1 ratio, model building of the nsp12-nsp7-nsp8 complex revealed unexpected density for a second subunit of nsp8 (a.a. 77–191) without a bound nsp7. This second subunit of nsp8 interacts with the nsp12 interface domain proximal to the fingers domain and the RNA template-binding channel. The identification of an interaction of nsp8 with the N-terminal region of nsp12 is supported by a recent study of infectious bronchitis virus, a *Gammacoronavirus*, where nsp12 a.a. 1–400 was sufficient to interact with nsp8[31]. The head domain of the second nsp8 subunit (a.a. 127–191) resembles the nsp8 head domains of previously determined crystal structures as well as the nsp7-nsp8 heterodimer bound to nsp12 discussed above (Fig. 4). However, the region of the second subunit of nsp8 a.a. 98–126 appears to adopt a conformation unique from those previously observed and involves a refolding of loops and a helical region to participate in interactions with nsp12 and precludes an interaction with nsp7.

SARS-CoV nsp7 and nsp8 have been extensively mutated and tested in in vitro polymerase activity assays[7]. Of the three nsp7 mutants previously found to diminish polymerase activity, the lysine of K7A hydrogen bonds to the carbonyls of the nsp7 25–42 helix that interacts with nsp12 and H36A and N37A mutations lie in the nsp7–nsp12 interface. Of the nsp8 residues previously mutated[7] and visible in the structure, four mutants were detrimental to polymerase function. Mutation of nsp8 P183A results in the mutation of a *cis*-proline residue and may result in an nsp8 folding defect. The three remaining mutated nsp8 residues make important contributions to interactions with nsp12 through the second nsp8 subunit, while these same nsp8 a.a. positions are distal to the observed interaction sites in the nsp7-nsp8 heterodimer. nsp8 D99A would disrupt an electrostatic interaction with nsp12 K332. P116 sits in a turn between two nsp8 helices and mutation to an alanine at this position may disrupt the ability of this loop to adopt the unique nsp12-bound conformation. Mutation of nsp8 R190A would disrupt the hydrogen bonding of R190 with carbonyls of nsp8 helix a.a. 116–121, which may destabilize the unique conformation of this

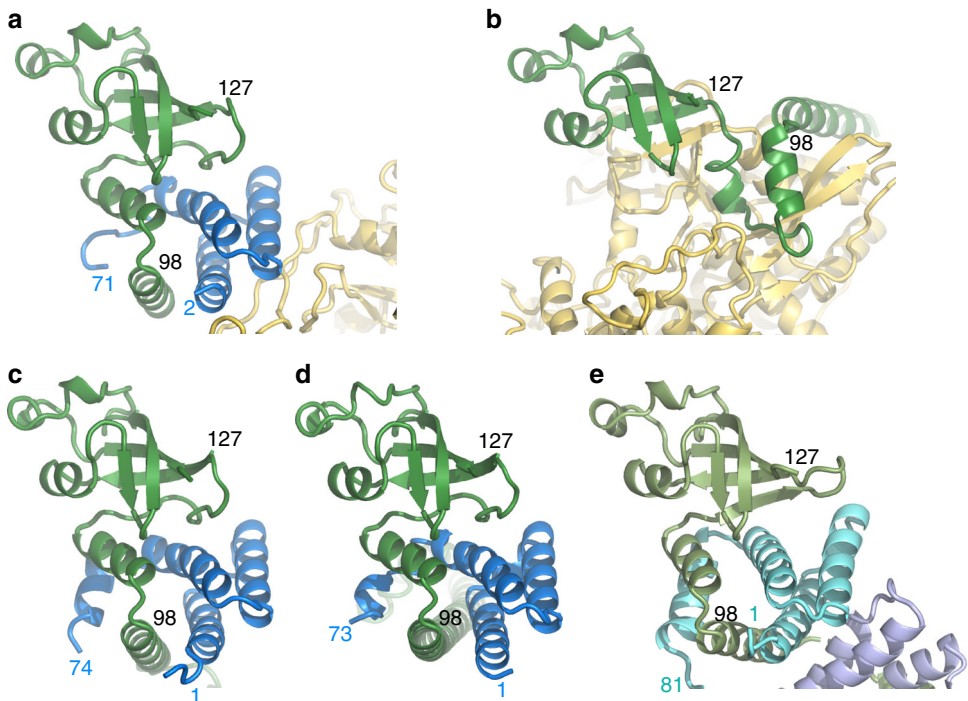

**Fig. 4** Comparison of nsp8 conformations. Two conformations of nsp8 are observed in the nsp12-nsp7-nsp8 structure. **a** A heterodimer of nsp7 (blue) and nsp8 (green) contacts nsp12 (yellow) primarily using surfaces of nsp7. **b** A second nsp8 subunit also contacts nsp12 directly using a unique conformation of nsp8 amino acids 98–127. The nsp7-nsp8 structures in two conformations for SARS-CoV[14] (2AHM.pdb) [10.2210/pdb2AHM/pdb] (**c**, **d**) as well as nsp8 (light green) and two subunits of nsp7 (cyan and purple) for Feline coronavirus[29] (3UB0.pdb) [10.2210/pdb3UB0/pdb] (**e**) show similar nsp8 conformations as observed in the SARS-CoV nsp7-nsp8 heterodimer bound to nsp12 (**a**)

nsp8 region and prevent nsp12 binding. These previous mutagenesis studies validate the protein interfaces presented here and emphasize the importance of the observed interactions of nsp12 with nsp7 and nsp8.

In three-dimensional (3D) classification of the SARS nsp12-nsp7-nsp8 complexes, we observed that a portion of the particles appeared to be missing the nsp7-nsp8 heterodimer (Supplementary Fig. 2). Further 3D sorting and refinement of these particles yielded a reconstruction at 3.5-Å resolution, which largely resembles the larger complex (Supplementary Fig. 5). However, in addition to missing the nsp7-nsp8 heterodimer, this structure also displays a significant amount of disorder in the fingers domain index and ring loops, which play a role in template and NTP binding (Fig. 5). We hypothesize that the binding of the nsp7-nsp8 heterodimer to the index finger loop stabilizes the polymerase domain to permit template recognition. This hypothesis is consistent with the observation that, while nsp12, nsp7, and nsp8 all individually recognize RNA with no or poor affinity, the complex of nsp12-nsp7-nsp8 is significantly more capable of binding RNA[7].

## Discussion
Here we have described the 3.1 Å cryo-EM structure of the SARS-CoV nsp12 polymerase bound to nsp7 and nsp8 co-factors resolving 123 kDa of the 160-kDa complex. The NiRAN domain of nsp12 presents a conserved face for interaction with the N-terminal disordered domain. Structural homology of nsp12 to the polymerases of the picornavirus family suggests residues involved in template recognition and catalysis. The role of nsp7 and nsp8 heterodimers appears to be the stabilization of nsp12 regions involved in RNA binding while a second subunit of nsp8 also plays a crucial role in polymerase activity possibly by extending the template RNA-binding surface.

Although the SARS-CoV polymerase has been found to be capable of carrying out de novo RNA initiation, the CoV polymerase structure lacks a clear indication for how this initiation occurs. Structurally, the SARS-CoV nsp12 most closely resembles the polymerases of picornaviruses, such as poliovirus, norovirus, and rhinovirus[24,25]. Picornaviruses use a protein primer called VPg to prime RNA synthesis and serve as a 5′ cap. However, CoVs cap their RNAs with a 5′ GTP resembling host RNAs[33] and hence are likely to use a distinct mechanism for RNA initiation. Biochemical evidence has shown that the nsp12-nsp7-nsp8 complex possesses de novo initiation activity that is dependent on the nsp12 polymerase-active site[7] similar to NS5 and NS5B polymerases of the *Flaviviridae* family such as dengue virus and hepatitis C virus, which use protein elements within the thumb domain to position a priming nucleotide in the −1 primer position[34]. However, the SARS-CoV nsp12 double-stranded RNA exit tunnel has no obstructions to act as a platform for priming nucleotides. Hence, the mechanism by which CoV polymerases carry out de novo RNA initiation is likely to be distinct from either the picornaviruses or flaviviruses, but the nature of this mechanism remains unclear.

A search for structural homology with SARS-CoV nsp12 NiRAN domain using the DALI server[35] revealed significant structural homology to protein kinases (Supplementary Table 2 and Supplementary Fig. 6). The homologous region spans the C-terminal large kinase domain, with the closest structural homology residing in the kinase nucleotide-binding site. Though kinase activity has not been demonstrated for nsp12, there are several shared kinase catalytic residues visible in the SARS-CoV nsp12 NiRAN structure including Asn209 and the nidovirus-conserved Asp218 and Phe219. As the nsp12 N-terminus is disordered, we are unable to determine whether this region of nsp12 resembles the small kinase domain. However, the shared residue

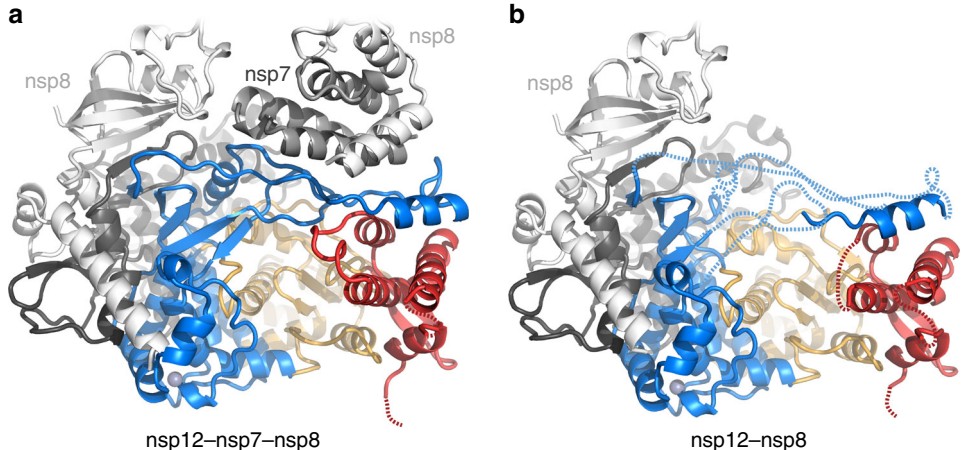

**Fig. 5** The nsp7-nsp8 heterodimer stabilizes loops of the fingers domain. **a** The nsp7-nsp8 heterodimer binds nsp12 on the fingers domain (blue) index finger loop adjacent to the palm (yellow) thumb (red) domains. **b** In the absence of the nsp7-nsp8 heterodimer, the fingers domain index and ring finger loops as well as several connecting loops in the thumb domain become disordered (delineated by dashed lines). The nsp12 N-terminal extension, nsp7, and nsp8 are shown in black, gray, and white, respectively

identities with the large kinase domain do suggest a potential nucleotide-binding site within the nsp12 NiRAN domain. How such a kinase-like domain carries out the previously reported nucleotidyltransfer[9] remains a subject for further study. One possibility is a mechanism similar to that of the pseudokinase SelO, which binds ATP in an alternate configuration and possesses AMPylation activity[36,37].

Comparing both the nsp7-nsp8 heterodimer and the second subunit of nsp8 structures as bound to nsp12 with the previously determined SARS-CoV nsp7-nsp8 complex, we find the previously observed higher-order nsp7-nsp8 oligomer formation inferred from crystal packing to be incompatible with the interfaces observed in the nsp12-bound states. Consequently, a.a. identified as important for nsp8's RNA primase activity[15] would not be brought into close proximity. Hence, the structural data here is inconsistent with the formation of this proposed primase-active site but is in agreement with previous de novo initiation experiments of the nsp7-nsp8-nsp12 complex demonstrating a lack of primase activity for nsp8 and de novo initiation activity for the nsp12 polymerase[7].

Viral co-factors nsp7 and nsp8 have been established as being essential for a highly active nsp12 polymerase complex. While the majority of nsp7 could be resolved in this structure, the N-terminal regions of both nsp8 subunits are disordered in agreement with previous observations and predictions[14,38]. Given the large number of other viral nsps and RNA that have been assigned to contact nsp8, we speculate that these disordered N-terminal regions may act as molecular handles for recruiting additional viral factors and organizing the viral replication complex.

The CoV nsp12 polymerase structure, presented here, fills a gap in our current understanding of CoV non-structural proteins involved in RNA synthesis. These viral nsps form a large multi-subunit complex, but how this complex is assembled around the key RNA polymerase-active site has until now remained unclear. This model of nsp12 bound to nsp7 and nsp8 is the first step toward unraveling the assembly and activity of this viral macromolecular complex with key implications for future studies of CoV antiviral design.

## Methods
**Protein production**. The gene for SARS-CoV nsp12 encompassing the 931 N-terminal a.a. was chemically synthesized with codon optimization (Genscript).

The −1 ribosomal frameshifting that naturally occurs to produce nsp12 was corrected to express the nsp12 open reading frame without frameshifting (Supplementary Table 3). The gene was cloned (Supplementary Table 4) into a modified pFastBac vector containing a 5′ Kozak sequence and C-terminal Thrombin protease site, hexahistidine, and Strep tags using the NEB HiFi DNA Assembly Kit (New England Biolabs). This plasmid construction adds additional a.a. MG to the N terminus of the protein and LVPRGSGHHHHHHAWSHPQFEK to the C terminus. This transfer plasmid was used to create recombinant bacmids by transformation into *Escherichia coli* DH10Bac (Life Technologies). The nsp12 bacmid was transfected into Sf9 cells (Expression Systems) using Cellfectin II (Life Technologies). The recombinant baculovirus was amplified twice in Sf9 cells. Thirteen milliliters of the second amplification was used to infect 1 L of Sf21 cells (Expression Systems) at $2.8 \times 10^6$ cells/mL and incubated at 27 °C for 48 h. Cells were harvested by centrifugation at $1000 \times g$ and resuspended in 70 mL of 25 mM HEPES pH 7.4, 300 mM sodium chloride, 1 mM magnesium chloride, and 5 mM beta-mercaptoethanol. The resuspended cells were then mixed with an equal volume of the same buffer containing 0.2% (v/v) Igepal CA-630 (Anatrace) and incubated with agitation for 10 min at 4 °C. The cell lystate was then sonicated before clarification by centrifugation at $35,000 \times g$ for 30 min and passing through a 0.45-μm filter. Cleared lysates were loaded onto Streptactin Agarose (Qiagen), washed with same buffer, and eluted with same buffer containing 2 mM desthiobiotin. Eluted protein was concentrated and further purified by size exclusion chromatography using a Superdex200 column (GE Life Sciences) in 25 mM HEPES pH 7.5, 300 mM NaCl, 0.1 mM magnesium chloride, and 2 mM tris(2-carboxyethyl)phosphine. Fractions containing nsp12 were concentrated with an Amicon Ultra concentrator (Millipore Sigma).

Full-length, chemically synthesized, and codon-optimized nsp7 and nsp8 genes (Supplementary Table 3) were cloned into pET46 (Supplementary Table 4) for expression in *E. coli* using Ek/LIC cloning (Novagen). While the sequence encoding the TEV protease site was cloned into pET46 when inserting nsp7, the TEV protease site for nsp8 was added in a second step using insertional mutagenesis. The N-terminal tags for nsp7 are MAHHHHHHVDDDDKMENLYFQG and for nsp8 are MAHHHHHHVDDDDKMENLYFQ. The TEV protease cleavage sites (ENLYFQ|G) were positioned to leave an N-terminal glycine on nsp7 and no additional N-terminal residues on nsp8. Plasmids were transformed into Rosetta2 pLysS *E. coli* (Novagen). Bacterial cultures were grown to an OD$_{600}$ of 0.8 at 37 °C, and then the expression was induced with a final concentration of 1 mM of isopropyl β-D-1-thiogalactopyranoside and the growth temperature was reduced to 16 °C for 14–16 h. Cells were harvested by centrifugation and were resuspended in 10 mM HEPES pH 7.4, 300 mM sodium chloride, 30 mM imidazole, and 5 mM beta-mercaptoethanol. Resuspended cells were lysed using a cell disruptor (Constant Systems) operating at 20,000 PSI. Lysates were cleared by centrifugation at $35,000 \times g$ for 30 min and then filtration using a 0.45-μm vacuum filter. Lysates were bound to Ni-NTA agarose (Qiagen), washed with same buffer, and then protein was eluted with same buffer containing 300 mM imidazole. Eluted proteins were digested with 1% (w/w) TEV protease at room temperature overnight while dialyzing into 10 mM HEPES pH 7.4, 300 mM sodium chloride, and 5 mM beta-mercaptoethanol. TEV protease-digested proteins were passed over Ni-NTA to remove uncleaved proteins and then further purified by size exclusion chromatography using a Superdex200 column (GE Life Sciences) in 25 mM HEPES pH 7.4, 300 mM sodium chloride, 0.1 mM magnesium chloride, and 2 mM tris (2-carboxyethyl)phosphine. Fractions containing the purified proteins were concentrated using Amicon Ultra concentrators (Millipore Sigma).

**Electron microscopy and model building**. Purified nsp12 was combined with nsp7 and nsp8 in a 1:2.2:2.2 molar ratio and incubated at 4 °C overnight. Three microliters of the protein complex was combined with 0.5 μL of 0.42 mM n-dodecyl-β-D-maltopyranoside and then immediately spotted onto UltraAuFoil 300 mesh 1.2/1.3 holey grids that had been plasma cleaned for 7 s with an Ar/O$_2$ gas mix. Grids were immediately blotted for 3 s using a Vitrobot (Thermo Fisher) and then plunge frozen in liquid ethane. EM data were collected using a Talos Arctica (Thermo Fisher) operating at 200 kV with a Gatan K2 detector[39]. A dose rate of 5.7 $e^-$/pix/s was used with a 11.75-s exposure for a total dose of 50.5 $e^-$/Å$^2$, which was fractionated across 47 movie frames (250 ms/frame). One thousand six hundred and seventy-seven micrographic movies were collected using image shift and a defocus range of 0.4–1.0 μm using the Leginon data collection software[40]. Movie frames were aligned using MotionCor2[41]. All 47 movie frames were used to produce the aligned images. CTF parameters were estimated from the aligned micrographs using GCTF[42]. Particles were picked using DoG picker[43] and filtered after manual image assessment using EMHP[44]. Particles were extracted using RELION-3.0[45] and several rounds of two-dimensional classification was performed in Cryosparc v0.6.5[46]. Clean stacks were used to reconstruct and refine an SGD initial model using RELION-3.0. 3D classification was performed in RELION-3.0 to identify subsets of particles giving rise to a high-resolution reconstruction and the best subsets was subsequently re-refined in RELION-3.0 (Supplementary Fig. 2).

The initial coordinates of rhinovirus serotype 16 polymerase[47] (1XR5.pdb) [10.2210/pdb1XR5/pdb] and SARS-CoV nsp7 and nsp8[14] (2AHM.pdb) [10.2210/pdb2AHM/pdb] were rigid body docked into the map and used as starting models for coordinate modeling. Rebuilding and sequence assignment was performed using Coot[48]. The coordinate models were refined using RosettaRelax[49] and phenix.real_space_refine[50]. Final models were assessed using Molprobity[51] and EMRinger[52].

**Sequence alignment**. nsp12 protein sequences were derived from available CoV ORF1ab sequences. These sequences are comprised of HKU19 (YP_005352862.1), porcine deltacoronavirus (AMN91620.1), HKU11 (YP_002308478.1), transmissible gastroenteritis virus (P0C6Y5.1), 229E-related bat CoV (APD51497.1), HuCoV-NL63 (AVA26872.1), porcine epidemic diarrhea virus (AKJ21970.1), HKU8 (YP_001718610.1), HKU22 (AHB63507.1), infectious bronchitis virus (AAS00078.1), turkey CoV (ACV87277.1), HuCoV-OC43 (YP_009555238.1), murine hepatitis virus (YP_209229.2), HuCoV-HKU1 (ARB07596.1), MERS-CoV (ATG84853.1), HKU4 (ABN10847.1), SARS-CoV (AAP33696.1), and HKU9 (AVP25405.1). These sequences were aligned with Clustal Omega[53]. A subset of these aligned sequences is shown in Supplementary Fig. 3. The full alignment was used as input for ConSurf[54] to generate the images in Fig. 2.

**nsp12 activity assay**. The activity of SARS-CoV nsp12 polymerase was tested using a modification of the primer extension assay previously described[7]. A 40-nt template RNA corresponding to the 3′ end of the SARS-CoV genome was annealed to a complementary 20-nt primer containing a 5′ fluorescein label. Primer extension reactions were assembled containing 500 nM nsp12, 1.5 μM nsp7, 1.5 μM nsp8, 250 nM annealed primer/template with 10 mM Tris HCl pH 8.0, 10 mM potassium chloride, 2 mM magnesium chloride, and 1 mM beta-mercaptoethanol. Reactions were incubated at room temperature for 15 min before the addition of an NTP mix (500 μM each, final concentration). Reactions were incubated at 30 °C for 60 min and then analyzed by denaturing TBE-Urea PAGE with fluorescence detection (ChemiDoc MP, BioRad).

**Reporting summary**. Further information on research design is available in the Nature Research Reporting Summary linked to this article.

## Data availability

Reconstructed density maps and refined coordinates have been deposited in the Electron Microscopy Database (EMDB-0520 and EMDB-0521) and the Protein Databank (6NUR and 6NUS). Other data are available from the corresponding author upon reasonable request.

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

# ARTICLE

31. complex independent of the non-translated regions of viral RNA and other viral proteins. *Virology* **513**, 75–84 (2018).
32. von Brunn, A. et al. Analysis of intraviral protein-protein interactions of the SARS coronavirus ORFeome. *PLoS ONE* **2**, e459 (2007).
33. Chen, Y. & Guo, D. Molecular mechanisms of coronavirus RNA capping and methylation. *Virol. Sin.* **31**, 3–11 (2016).
34. Choi, K. H., Rossmann, M. G. & RNA-dependent RNA polymerases from Flaviviridae. *Curr. Opin. Struct. Biol.* **19**, 746–751 (2009).
35. Holm, L. & Laakso, L. M. Dali server update. *Nucleic Acids Res.* **44**, W351–W355 (2016).
36. Kuchibhatla, D. B. et al. Powerful sequence similarity search methods and in-depth manual analyses can identify remote homologs in many apparently "orphan" viral proteins. *J. Virol.* **88**, 10–20 (2014).
37. Sreelatha, A. et al. Protein AMPylation by an evolutionarily conserved pseudokinase. *Cell* **175**, 809–821.e819 (2018).
38. Sutton, G. et al. The nsp9 replicase protein of SARS-coronavirus, structure and functional insights. *Structure* **12**, 341–353 (2004).
39. Herzik, M. A. Jr., Wu, M. & Lander, G. C. Achieving better-than-3-A resolution by single-particle cryo-EM at 200 keV. *Nat. Methods* **14**, 1075–1078 (2017).
40. Suloway, C. et al. Automated molecular microscopy: the new Leginon system. *J. Struct. Biol.* **151**, 41–60 (2005).
41. Zheng, S. Q. et al. MotionCor2: anisotropic correction of beam-induced motion for improved cryo-electron microscopy. *Nat. Methods* **14**, 331–332 (2017).
42. Zhang, K. Gctf: Real-time CTF determination and correction. *J. Struct. Biol.* **193**, 1–12 (2016).
43. Voss, N. R., Yoshioka, C. K., Radermacher, M., Potter, C. S. & Carragher, B. DoG Picker and TiltPicker: software tools to facilitate particle selection in single particle electron microscopy. *J. Struct. Biol.* **166**, 205–213 (2009).
44. Berndsen, Z., Bowman, C., Jang, H. & Ward, A. B. EMHP: an accurate automated hole masking algorithm for single-particle cryo-EM image processing. *Bioinformatics* **33**, 3824–3826 (2017).
45. Zivanov, J. et al. New tools for automated high-resolution cryo-EM structure determination in RELION-3. *Elife* **7**, e42166 (2018).
46. Punjani, A., Rubinstein, J. L., Fleet, D. J. & Brubaker, M. A. cryoSPARC: algorithms for rapid unsupervised cryo-EM structure determination. *Nat. Methods* **14**, 290–296 (2017).
47. Love, R. A. et al. The crystal structure of the RNA-dependent RNA polymerase from human rhinovirus: a dual function target for common cold antiviral therapy. *Structure* **12**, 1533–1544 (2004).
48. Emsley, P., Lohkamp, B., Scott, W. G. & Cowtan, K. Features and development of Coot. *Acta Crystallogr. D Biol. Crystallogr.* **66**, 486–501 (2010).
49. DiMaio, F. et al. Atomic-accuracy models from 4.5-A cryo-electron microscopy data with density-guided iterative local refinement. *Nat. Methods* **12**, 361–365 (2015).
50. Adams, P. D. et al. PHENIX: a comprehensive Python-based system for macromolecular structure solution. *Acta Crystallogr. D Biol. Crystallogr.* **66**, 213–221 (2010).
51. Williams, C. J. et al. MolProbity: more and better reference data for improved all-atom structure validation. *Protein Sci.* **27**, 293–315 (2018).
52. Barad, B. A. et al. EMRinger: side chain-directed model and map validation for 3D cryo-electron microscopy. *Nat. Methods* **12**, 943–946 (2015).
53. Sievers, F. & Higgins, D. G. Clustal Omega for making accurate alignments of many protein sequences. *Protein Sci.* **27**, 135–145 (2018).
54. Ashkenazy, H. et al. ConSurf 2016: an improved methodology to estimate and visualize evolutionary conservation in macromolecules. *Nucleic Acids Res.* **44**, W344–W350 (2016).
55. Zhao, Y. et al. A crystal structure of the Dengue virus NS5 protein reveals a novel inter-domain interface essential for protein flexibility and virus replication. *PLoS Pathog.* **11**, e1004682 (2015).

## Acknowledgements

We gratefully acknowledge help and advice on electron microscope alignment and data processing from Mark A. Herzik, Bill Anderson, and Hannah L. Turner. We also thank Charles A. Bowman and Jean-Christopher Ducom for computational support and to Lauren G. Holden for a critical reading of this manuscript. Thanks also to David Karlin for pointing us to the pseudokinase SelO as a potential functional homolog of the nsp12 NiRAN domain. This work was funded by the National Institute for Allergy and Infectious Disease AI123498 to R.N.K. and AI127521 to A.B.W.

## Author contributions

Both authors designed experiments, wrote and edited the manuscript. R.N.K expressed and purified protein, prepared EM grids, collected and processed EM data, and tested protein activity.

## Additional information

**Competing interests:** The authors declare no competing interests.

