## [Peer Review File · Nature Communications]

Reviewers' Comments:

Reviewer #1:

Remarks to the Author:

Kirchdoerfer et al. report the cryo-EM structure of the SARS-CoV NSP12 in complex with its co-factors NSP7 and NSP8 at 3.1 angstrom resolution. Their findings are: 1) a NSP12 N-terminal NiRAN domain structurally homologous to protein kinases; 2) a conserved NSP12 RNA-dependent RNA polymerase domain; 3) in addition to the NSP7-NSP8 heterodimer binding to the NSP12, there is another copy of NSP8 occupying a distinct binding site on the NSP12.

Here is a list of my comments:

Major points:

P11, L199. The authors suggested that the NSP12-NSP7-NSP8 complex possesses de novo initiation activity similar to HCV polymerase. Did the authors compare the NSP12 structure with HCV (+ssRNA virus) polymerase structures in primed initiation and elongation states (See Appleby et. al., Science 2015, V.347, p.771)?

P12, L201-204. This maybe too speculative. The authors cannot exclude the possibility that the NSP12 has other conformations in different states, similar to the HCV polymerase. In addition, nsp12 activity was previously suggested to be primer dependent (Nucleic Acids Research, V.38, 2010, P.203–214), would the authors like to comment on this?

P12, L215-223. This paragraph is not clear to me. The background of GS-5734 is not presented.

P13, L235-237. A brief functional definition of the primase might be given here or in the introduction part. Could the authors exclude the possibility that the NSP12 initiation is primer dependent? See comment above.

P14, L247. Do the authors mean the NSP7-NSP8 on the NSP12?

P15, L282. Could the authors briefly describe the results of the biochemical studies here?

Minor point:

Supplemental/Supplemental/Supplemental and Fig/Fig./Figure are used in the main text and the supplementary information.

Fig.1

Compared with the middle panel of Fig.1b, it seems the grey structures (upper left and upper right) in Fig. 1c are NSP8 and NSP7, not the N-terminal extensions of NSP12 as described in the figure legend.

P9, L140-142. Limited flexibility does not affect a low resolution reconstruction. To see if the N-terminal structure is visible, the authors could try to filter to a lower resolution.

Fig.4

lowercase letters in the panels.

P13, L245-246. This sentence is not clear.

Fig.5

The color scheme for the subunits is not clear.

P20, L399. Were all 47 frames aligned and averaged to a single image for the final reconstruction?

Reviewer #2:

Remarks to the Author:

Kirchdoerfer and Ward presented EM structure of a multiprotein complex including nsp7, nsp8, and nsp12 of SARS-CoV, which is part of replication-transcription complex (RTC) of nidoviruses. While structure of nsp7 and nsp8 was elucidated before, other aspects of this study, including the structure of nsp12 and interaction of this protein with nsp7 and nsp8 are novel. Significance of these findings is difficult to overestimate: nsp12 includes RdRp domain that is arguably the most important enzyme of replication; it is the largest core component of RTC and the last which structure has been solved. It is not due to the lack of effort by many fine groups that it took 30 years since nsp12 and RdRp were identified (PMID: 2842734 and 2526320) before this study succeeded. Congratulations to the authors!

The presented structure is in good agreement with results of numerous structural, biochemical, molgenetics, and bioinformatics studies involving different coronaviruses. Its principal novelties concern elucidation of structures of the N-terminal part of nsp12 adjacent to the RdRp domain, which includes NiRAN domain, and interfaces between nsp12 and two small proteins, nsp7 and nsp8. Interaction between RdRp and others is central to understanding of how replication and transcription of giant RNA genomes of coronaviruses and other nidoviruses are regulated. Findings that NiRAN adopts a kinase-like domain and nsp8 may interact with nsp12 in two different conformations, with and without nsp7 involved, are particularly intriguing. The solved structure forms a platform to advance our still rudimentary knowledge of this complex process, also along new directions of research.

I have no major concerns, although wish to have seen more details concerning expression of the analysed proteins and identification of the kinase-like domain in nsp12. Could technical issues be excluded as contributing to the disordered state of the very N-terminus of nsp12 and to the nsp7-independent interaction of nsp8 with nsp12?

Specific suggestions:

Nidovirologists agreed to use "nsp" rather than "NSP" for non-structural proteins.

Allocation of text to different sections could be improved as detailed below in several places.

16: revise this sentence, since no expectation was defined, e.g. "...fold is bound by two nsp8 co-factors in different conformations".

20-35. Shorten this part to retain background relevant to this study.

58: Replace "unexpected" with "peculiar" or alike, since no expectation was defined.

84: Replace "nidovirus-unique extension" with "NiRAN" to clarify location

84-88 and Supp. Fig. 3: the selected viruses represent only a fraction of the known coronavirus diversity, with delta coronaviruses being the most notable omission. Some of the listed residues may not be conserved in these viruses.

125-137 and 226-246. Relocate these pieces of text about the domain and functional complexity of nsp12 and controversy over the function and structure of nsp7-nsp8 from the Results to the Introduction. These details help the reader appreciate motivation and challenges of the study right from the beginning, and introduce NiRAN domain which is part of Fig. 1 but was mentioned first time in the current text only after Fig. 2 was described.

132: Replace "NTPs" with "GTP/UTP"

133: Insert 'and virus' after "activity"

157 and elsewhere: use italic for taxa names

173-174. Relocate this sentence from the Results to the Introduction to introduce and cite Fig. 1A properly.

192-204. Move to the Discussion.

192: clarify that the problem is with our understanding of the mechanism used rather than with the mechanism itself. Consult Discussion in Ref. 10 about possible options.

199: Insert "NS5/NS5B of " before "similar"

213: Insert " motif A" upstream of Asp623.

214: Is conservation of these residues exclusive to coronaviruses? Otherwise, it may be not a decisive factor in the resistance to antiviral nucleoside analogs. Please clarify.

268-271. Move to the Discussion.

273: Cite Fig. 4C-E

276-283: Move to the Discussion

284-299: All discussed residues and their interactions must be referred to Figure(s) where they highlighted. Consider moving this part to the Discussion.

288: Change "...structure, four..." to "...structure, mutations of four..." or revise the entire sentence to improve its clarity.

290-291: Replace "positions" with "residues"

297-299: Revise sentence along this line "The interprotein interfaces in the reported structure of the nsp7-8-12 complex rationalize results of mutagenesis studies and emphasize the ..."

326-328. Details of DALI use and statistics of similarity between NiRAN and kinases should be documented. Providing a Supp Table with the hit list and statistics could be informative.

329-332: Poor phrasing, revise. If the authors want to propose kinase activity for NiRAN, catalytic residues of kinases and NiRAN motifs should be compared. Otherwise, "overlapping kinase residues" is confusing statement.

353-354. Details of the construction used along with Refs, if necessary, should be included. Particularly important to detail is how the authors dealt with the ORF1a/1b ribosomal frameshifting site that interrupts nsp12 locus in the genome.

371-372. Details of the constructions used along with Refs, if necessary, should be included.

Fig. 1A. Consider including a schematic of nsp7 and nsp8 on top of that for nsp12 to include details mentioned in the text.

Fig. 2: define all colors used

Fig. 4: define all colors used

Suppl. Fig. 3. Adding secondary structure elements to the sequence alignment will enrich this figure.

Suppl. Fig. 6 legend. Replace "GDP" with "ADP".

Reviewer 1:

P11, L199. The authors suggested that the NSP12-NSP7-NSP8 complex possesses *de novo* initiation activity similar to HCV polymerase. Did the authors compare the NSP12 structure with HCV (+ssRNA virus) polymerase structures in primed initiation and elongation states (See Appleby et. al., Science 2015, V.347, p.771)?

Comparison of SARS-CoV nsp12 to HCV NS5B initiation and elongation structures suggests that dissimilar to HCV, SARS-CoV does not contain a β -loop-like structure to create a platform for incoming nucleotides during *de novo* initiation nor is there evidence that the C-terminus of the nsp12 polymerase inserts into the primer/template exit channel.

P12, L201-204. This maybe too speculative. The authors cannot exclude the possibility that the NSP12 has other conformations in different states, similar to the HCV polymerase. In addition, nsp12 activity was previously suggested to be primer dependent (Nucleic Acids Research, V.38, 2010, P.203–214), would the authors like to comment on this?

The wording of this sentence indicates that the existence of a distinct initiation mechanism is a hypothesis based on available data. We believe it would be too speculative to suggest that alternate conformational states of nsp12 exist as comparison to numerous picornavirus polymerases which are closer structural homologues appears to indicate the opposite. The conclusion of primer dependent activity previously proposed was based on the weak activity of nsp12 polymerase alone acting on specialized homopolymeric RNA substrates (te Velthuis et al. Nuc Acid Res 2010). rather than reflecting the activity of the nsp12-nsp7-nsp8 complex which has greater activity and works on more biologically relevant substrates (Subissi et al. PNAS 2014).

P12, L215-223. This paragraph is not clear to me. The background of GS-5734 is not presented.

GS-5734 is a nucleoside analog which targets the viral RNA synthesis machinery acting either by premature chain termination or lethal mutagenesis. Resistance mutations to this analog indicate amino acids involved in modulating polymerase fidelity such as Valine557. Clarifying statements have been made in the text.

P13, L235-237. A brief functional definition of the primase might be given here or in the introduction part. Could the authors exclude the possibility that the NSP12 initiation is primer dependent? See comment above.

A description of the primase activity has been included. Several publications have shown that preparations of nsp8 are capable of synthesizing short RNA oligonucleotides. However, the nsp8 primase activity is controversial and it is difficult to reconcile these results with expected mutagenesis and structural data. The assertion that nsp12 is primer independent is taken from Subissi et al. who showed that a nsp12-nsp7-nsp8 complex is capable of *de novo* initiation and that this activity was dependent on a functional nsp12 polymerase active site.

P14, L247. Do the authors mean the NSP7-NSP8 on the NSP12?

Yes. A clarification has been made to the text

P15, L282. Could the authors briefly describe the results of the biochemical studies here?

The previous biochemical studies comprise experiments on the *de novo* initiation of the nsp7-nsp8-nsp12 complex showing that nsp8 did not synthesize short oligonucleotides

as part of the nsp7-nsp8-nsp12 complex and that *de novo* initiation was dependent on the nsp12 polymerase active site. Statements describing this previous work has been included in the text.

Supplemental/Supplemental/Supplemental and Fig/Fig./Figure are used in the main text and the supplementary information.

Figure and supplementary figure callouts have been standardized for Nature Communications.

Fig.1

Compared with the middle panel of Fig.1b, it seems the grey structures (upper left and upper right) in Fig. 1c are NSP8 and NSP7, not the N-terminal extensions of NSP12 as described in the figure legend.

The figure and legend have been modified to be clearer.

P9, L140-142. Limited flexibility does not affect a low resolution reconstruction. To see if the N-terminal structure is visible, the authors could try to filter to a lower resolution.

While it is true that domains with limited flexibility can be seen at low resolution, larger motions average out in EM reconstructions. The nsp12-nsp7-nsp8 reconstruction and low pass filtered maps (below) do indicated density for the NiRAN domain, but this density is broken and blobby precluding accurate assignment of the N-terminal 116 amino acids and indicating a large degree of domain motion.

Fig.4

lowercase letters in the panels.

The figure has been updated

P13, L245-246. This sentence is not clear.

The statement has been clarified

Fig.5

The color scheme for the subunits is not clear.

The coloring scheme is the same as presented in Fig. 1b. The figure legend has been clarified.

P20, L399. Were all 47 frames aligned and averaged to a single image for the final reconstruction?

Yes, all 47 frames were aligned prior to particle extraction. The text has been clarified.

Reviewer 2:

Could technical issues be excluded as contributing to the disordered state of the very N-terminus of nsp12 and to the nsp7-independent interaction of nsp8 with nsp12?

It is difficult to exclude every technical issue as contributing to the disordered nsp12 N-terminus. The EM reconstruction appears to be representative of the prepared sample. The nsp7-independent interaction of nsp8 with nsp12 appears to be more stable than the interaction of the nsp7-nsp8 heterodimer with nsp12 as no nsp12 particles were identified that lacked the single nsp8 while a significant proportion lacked the nsp7-nsp8 heterodimer (Supplementary Fig. 2). The interaction of the single nsp8 is also supported by the cited mutagenesis and interaction studies (Tan et al. 2018, Subissi et al. 2014) suggesting a strong biological relevance.

Nidovirologists agreed to use “nsp” rather than “NSP” for non-structural proteins.

We have changed all instances of NSP to nsp

16: revise this sentence, since no expectation was defined, e.g. “...fold is bound by two nsp8 co-factors in different conformations”.

The word ‘unexpectedly’ was removed

20-35. Shorten this part to retain background relevant to this study.

This paragraph has been edited for brevity.

58: Replace “unexpected” with “peculiar” or alike, since no expectation was defined.

An expectation for nsp8 to exist as a 1:1 complex with nsp7 has been included in the results section.

84: Replace “nidovirus-unique extension” with “NiRAN” to clarify location

As described in the manuscript, we use nidovirus-unique extension and NiRAN differently to indicate different regions of the protein. We use nidovirus-unique extension to refer to all residues 1-397 which includes the NiRAN (a.a. 1-250) as well as the interface domain (a.a. 251-397). The text has been clarified to better indicate this.

84-88 and Supp. Fig. 3: the selected viruses represent only a fraction of the known coronavirus diversity, with delta coronaviruses being the most notable omission. Some of the listed residues may not be conserved in these viruses.

Porcine deltacoronavirus is listed as the first sequence in the Supplementary Figure 3 alignment. We based this statement on a much larger multiple sequence alignment which included deltacoronaviruses (HKU19, PDCV and HKU11), alphacoronaviruses (TGEV, 229E, NL63, PEDV, and HKU8), gammacoronaviruses (HKU22, IBV, and Turkey coronavirus) and betacoronaviruses (OC43, MHV, HKU1, MERS, HKU4, SARS and HKU9). In all of these sequences, the indicated zinc-coordinating residues are completely conserved. While these sequences do not represent to total diversity of the coronavirus family, the inclusion of diverse family members from all four genera is sufficient to say that

the residues are highly conserved. Although the full alignment could be included in the supplemental material, it is very large and adds little to the conclusions drawn. Hence, we instead opted to include representatives of all four genera with special emphasis on the betacoronaviruses which contain important human pathogens. A list of the sequences and accession numbers used for alignment have been added to the methods section.

125-137 and 226-246. Relocate these pieces of text about the domain and functional complexity of nsp12 and controversy over the function and structure of nsp7-nsp8 from the Results to the Introduction. These details help the reader appreciate motivation and challenges of the study right from the beginning, and introduce NiRAN domain which is part of Fig. 1 but was mentioned first time in the current text only after Fig. 2 was described. Sentences have been added to the introduction introducing the nucleotidyltransferase activity of nsp12 and some of the unknowns involving nsp7-nsp8. We have left the indicated sentences in the results sections to best introduce readers to these topics prior to the presentation of these specific structural features.

132: Replace “NTPs” with “GTP/UTP”
Replacement made

133: Insert ‘and virus’ after “activity”
‘and viral growth and recovery’ has been added.

157 and elsewhere: use italic for taxa names
Italics for taxa names have been included.

173-174. Relocate this sentence from the Results to the Introduction to introduce and cite Fig. 1A properly.
This background sentence introduces observations of this region of the protein. Though it is background, it’s placement here is to introduce the polymerase active site before a more detailed discussion of the results.

192-204. Move to the Discussion.
Moved to discussion

192: clarify that the problem is with our understanding of the mechanism used rather than with the mechanism itself. Consult Discussion in Ref. 10 about possible options.
The problem of de novo initiation has been clarified

199: Insert “NS5/NS5B of “ before “similar”
Included “NS5 and NS5B” to indicate a comparison to polymerases rather than viruses.

213: Insert “ motif A” upstream of Asp623.
Added motif A to this sentence.

214: Is conservation of these residues exclusive to coronaviruses? Otherwise, it may be not a decisive factor in the resistance to antiviral nucleoside analogs. Please clarify.
There is a general conservation of these residues across RNA polymerases though there appears to be pleiomorphism at Thr680. However, this portion of the text is meant to

describe the interrogation of incoming nucleotides, in particular the 2' and 3' hydroxyl positions. We have amended the text to make this clearer.

268-271. Move to the Discussion.

This is a citation for a previous result supporting the preceding observation and would lose meaning if taken out of its context within the results section.

273: Cite Fig. 4C-E

A citation for figure 4 has been added.

276-283: Move to the Discussion

Moved to discussion

284-299: All discussed residues and their interactions must be referred to Figure(s) where they highlighted. Consider moving this part to the Discussion.

Though it would be comprehensive to include such a figure, the residues discussed are widely spaced throughout the structure. This is especially difficult for nsp8 mutants as these would need to be represented twice to show positions in each copy of nsp8 observed in the structure. Showing each mutated residue discussed would require a large, multi-panel figure and after shrinking to fit journal requirements we believe such a figure would provide little information for readers. The mutations described in this paragraph are meant to support the observed interactions of nsp7 and nsp8 with nsp12 in the structure and shown in Fig. 1 and 4.

288: Change "...structure, four..." to "...structure, mutations of four..." or revise the entire sentence to improve its clarity.

The sentence has been clarified.

290-291: Replace "positions" with "residues"

Replaced with 'residues'

297-299: Revise sentence along this line "The interprotein interfaces in the reported structure of the nsp7-8-12 complex rationalize results of mutagenesis studies and emphasize the ..."

The suggested revision is stylistic and does not appear to provide additional clarity to this statement.

326-328. Details of DALI use and statistics of similarity between NiRAN and kinases should be documented. Providing a Supp Table with the hit list and statistics could be informative.

A hit list for the DALI search has been included in the supplement.

329-332: Poor phrasing, revise. If the authors want to propose kinase activity for NiRAN, catalytic residues of kinases and NiRAN motifs should be compared. Otherwise, "overlapping kinase residues" is confusing statement.

The suggestion that NiRAN has kinase activity is not proposed here or elsewhere in the manuscript. The word 'overlapping' has been changed to "shared" and refers to those residues in the kinase catalytic site that match those of the NiRAN after structure superposition.

353-354. Details of the construction used along with Refs, if necessary, should be included. Particularly important to detail is how the authors dealt with the ORF1a/1b ribosomal frameshifting site that interrupts nsp12 locus in the genome.

A complete description has been provided. The nsp12 gene was chemically synthesized and codon optimized to omit the necessity for the ribosomal frameshifting that occurs naturally in virus infection.

371-372. Details of the constructions used along with Refs, if necessary, should be included.

A complete description has been included.

Fig. 1A. Consider including a schematic of nsp7 and nsp8 on top of that for nsp12 to include details mentioned in the text.

Diagrams for nsp7 and nsp8 have been added.

Fig. 2: define all colors used

All colors defined.

Fig. 4: define all colors used

All colors defined.

Suppl. Fig. 3. Adding secondary structure elements to the sequence alignment will enrich this figure.

Secondary structure elements have been added.

Suppl. Fig. 6 legend. Replace "GDP" with "ADP".

The correction has been made.